# Adipocyte-Mediated Electrophysiological Remodeling of PKP-2 Mutant Human Pluripotent Stem Cell-Derived Cardiomyocytes

**DOI:** 10.3390/biomedicines12112601

**Published:** 2024-11-14

**Authors:** Justin Morrissette-McAlmon, Christianne J. Chua, Alexander Arking, Stanley Chun Ming Wu, Roald Teuben, Elaine Zhelan Chen, Leslie Tung, Kenneth R. Boheler

**Affiliations:** 1Department of Biomedical Engineering, Johns Hopkins University School of Medicine, Baltimore, MD 21205, USA; jmorri65@jhmi.edu (J.M.-M.); cchua3@jhmi.edu (C.J.C.); aarking4@jhmi.edu (A.A.); cwu116@jhu.edu (S.C.M.W.); rteuben1@jhmi.edu (R.T.); ltung@jhu.edu (L.T.); 2Department of Medicine, Division of Cardiology, Johns Hopkins University School of Medicine, Baltimore, MD 21205, USA; zchen119@jhmi.edu

**Keywords:** arrhythmogenic cardiomyopathy, human induced pluripotent stem cell-derived cardiomyocytes (hiPSC-CMs), adipocytes, electrophysiology, cytokines, plakophilin-2

## Abstract

Background: Arrhythmogenic cardiomyopathy (ACM) is a genetic disorder responsible for nearly a quarter of sports-related sudden cardiac deaths. ACM cases caused by mutations in desmosome proteins lead to right ventricular enlargement, the loss of cardiomyocytes, and fibrofatty tissue replacement, disrupting electrical and mechanical stability. It is currently unknown how paracrine factors secreted by infiltrating fatty tissues affect ACM cardiomyocyte electrophysiology. Methods: A normal and a PKP2 mutant (c.971_972InsT) ACM hiPSC line were cultivated and differentiated into cardiomyocytes (CMs). Adipocytes were differentiated from human adipose stem cells, and adipocyte conditioned medium (AdCM) was collected. Optical mapping and phenotypic analyses were conducted on human iPSC-cardiomyocytes (hiPSC-CMs) cultured in cardiac maintenance medium (CMM) and either with AdCM or specific cytokines. Results: Significant differences were observed in voltage parameters such as the action potential duration (APD_80_, APD_30_), conduction velocity (CV), and CV heterogeneity. When cultured in AdCM relative to CMM, the APD_80_ increased and the CV decreased significantly in both groups; however, the magnitudes of changes often differed significantly between 1 and 7 days of cultivation. Cytokine exposure (IL-6, IL-8, MCP-1, CFD) affected the APD and CV in both the normal and PKP2 mutant hiPSC-CMs, with opposite effects. NF-kB signaling was also found to differ between the normal and PKP2 mutant hiPSC-CMs in response to AdCM and IL-6. Conclusions: Our study shows that hiPSC-CMs from normal and mPKP2 ACM lines exhibit distinct molecular and functional responses to paracrine factors, with differences in RNA expression and electrophysiology. These different responses to paracrine factors may contribute to arrhythmogenic propensity.

## 1. Introduction

Arrhythmogenic cardiomyopathy (ACM) is generally an autosomal dominant genetic disorder with an incomplete penetrance and a population prevalence ranging from 1:1000 to 1:5000 [1]. ACM is often diagnosed in young adulthood, and it disproportionately affects males [2]. It is estimated to account for >10% of all cardiovascular deaths in patients younger than 65 years of age [3] and almost one quarter of all sports-related sudden cardiac deaths [4]. Approximately 30–50% of all cases of ACM and more than 60% of familial cases of ACM result from desmosome mutations [5,6]. Based on clinical and post-mortem pathological assessments of patient myocardium, desmosome-related ACM (dACM) is frequently associated with right ventricular enlargement, inflammation, the loss of cardiomyocytes, reduced contractility, and the replacement of the myocardium with fibrous and fatty tissue [7].

Cardiac desmosomes are essential for cell–cell contact and strong intercellular adhesion among adjacent cardiomyocytes. These structures comprise five proteins belonging to three gene families: the plakin family (desmoplakin), the cadherin superfamily (desmoglein-2, desmocollin-2), and the armadillo family (plakoglobin), which also includes members of the p120-catenin subfamily (plakophilin-2) [8]. The cadherin members of the desmosome complex link to intermediate filaments such as desmin through densely clustered cytoplasmic armadillo plaque proteins and plakin family cytolinkers [9]. Desmosomes also work cooperatively with adherens junctions, the anchoring sites for actin, and gap junctions to provide mechanoelectrical and metabolic coupling between adjacent cells [9]. These three molecular complexes work cooperatively between adjacent cardiomyocytes at the intercalated disk (ICD), subcellular structures that support the physical stability and integrity of heart tissues. Disruption to ICD structures can compromise cell–cell interactions, can perturb the electrical connectivity of the cardiac syncytium, and associate with a range of cardiomyopathies, including dACM [10]. 

Mutations in plakophilin-2 (PKP2) account for up to 60% of all cases of familial dACM [11,12]. PKP2, the primary plakophilin isoform in the heart, links cadherins to the intermediate filament cytoskeleton. It also regulates the signaling activity of β-catenin, which is known to control cell proliferation and differentiation and the transcription of genes, such as those associated with intracellular calcium cycling [13]. This protein is more prevalent in the right ventricle than the left. Patients with dACM caused by PKP2 mutations often exhibit right heart disease, though both ventricles and the intraventricular septum can be affected. Relative to patients with other desmosome mutations, patients with PKP2 mutations are often diagnosed at a younger age, exhibit more negative T waves, present with higher left ventricular ejection fractions, and are less likely to have symptoms of heart failure [11].

As dACM progresses, fibrous tissues typically intermingle with fat tissues. This fibrofatty infiltration contributes to right ventricular myocardial thinning by replacing cardiomyocytes typically in the apical right ventricular and subendocardial left ventricular tissue [14]. This progressive tissue remodeling has often been attributed to cardiomyocyte apoptosis or necrosis, which may contribute to persistent fibrosis, inflammation with the release of cytokines, and myocardial atrophy followed by myocardial replacement [15]. Desmosome-specific mutations also predispose the heart to reentrant electrical activity that is likely caused by macro-reentry around fibrofatty tissue deposits [1]. Nevertheless, the causes of fibrofatty infiltration and replacement in heart tissues and the associated changes in transmembrane voltages are not fully understood. Although several cellular and animal models of dACM are available to study ventricular dysfunction, as well as cardiomyocyte death and inflammation, fibrofatty infiltration processes characteristic of ACM have been difficult to recapitulate in vitro [16].

Human induced pluripotent stem cell-derived cardiomyocytes (hiPSC-CMs) represent an informative in vitro model for studying human cardiac diseases. For example, PKP2 mutant (mPKP2) hiPSC-CMs show disrupted desmosome localization, altered gap junction proteins and RNA levels, and reduced PKP2 RNA and protein levels [11]. Electrophysiologically, mPKP2 hiPSC-CMs demonstrate a decreased action potential upstroke velocity and prolonged field potential rise time. Furthermore, the addition of adipogenic factors to culture medium promotes the formation of lipid droplets in these cells, and these observations correspond with changes in the RNA abundances of adipogenic regulatory genes [17,18]. Recently, we reported the effects of adipocytes containing mostly white-type (energy-storing) or mixed/beige fat on the electrophysiology of normal hiPSC-CMs. We found that in vitro co-cultures of these adipocytes with cardiomyocytes generated from two normal hiPSC lines led to prolonged action potentials, reduced conduction velocities, and extended calcium transient durations. These electrophysiological changes were similarly observed when cardiomyocytes were co-cultured with adipocyte-conditioned medium or cultured in basal medium supplemented with cytokines, like MCP-1, that were secreted by these adipocytes [19]. Given the postulated disease-promoting effects of infiltrating fat during late stages and the increased incidence of the arrhythmias of patients caused by mPKP2 proteins, we have undertaken the current study to evaluate how paracrine factors may affect dACM cardiomyocyte electrophysiology. Specifically, we hypothesized that cytokines from adipocytes would cause changes in the electrophysiology of hiPSC-CMs from a mPKP2 line that might contribute to arrhythmia relative to normal CMs. To address this hypothesis, hiPSCs from a normal line and a mPKP2 ACM line were differentiated into cardiomyocytes and co-cultured with human adipocyte-conditioned medium or with medium supplemented with specific cytokines secreted by these adipocytes. The electrophysiology was assessed using the optical mapping techniques of monolayer cultures.

## 2. Materials and Methods

### 2.1. Human Induced Pluripotent Stem Cell Lines

One normal female hiPSC line (JHU001) and one male PKP2 mutant (mPKP2) ACM line (398-100) with a c.971_972InsT mutation [17] were used in this study in accordance with Johns Hopkins Medical Institutions ISCRO policies [18]. The cultivation of undifferentiated cells on Geltrex-coated 6-well plates in E8 medium was performed as previously described [19].

### 2.2. Cell Culture

Human iPSCs were differentiated as monolayers into cardiomyocytes (CMs) as previously described [19]. Undifferentiated hiPSC monolayers were maintained in E8 media (ThermoFisher Scientific Inc., Waltham, MA, USA) on 1:200 diluted Geltrex (ThermoFisher Scientific Inc., Waltham, MA, USA)-coated 6-well Nunc tissue culture plates (ThermoFisher Scientific Inc., Waltham, MA, USA). For cardiac differentiations, ~110,000 hiPSCs were seeded per well of a 6-well plate for four days [20] before the medium was switched to RPMI 1640 (ThermoFisher Scientific Inc., Waltham, MA, USA) containing a B27 supplement lacking insulin (ThermoFisher Scientific Inc., Waltham, MA, USA). This medium was supplemented with 6 µM CHIR99021 (Selleck Chemicals LLC, Houston, TX, USA) for 48 h from differentiation day 0 to differentiation day 2 (dd0–dd2). Following two days of culture, cells were re-fed with RPMI 1640 containing B27 minus insulin for 24 h (dd2–dd3) and then treated with 5 µM IWR-1 (MilliporeSigma, Burlington, MA, USA) in the same medium for 48 h (dd3–dd5). On dd8 or dd9, RPMI 1640 containing B27 minus insulin medium was switched to cardiomyocyte maintenance medium (CMM) consisting of RPMI 1640 supplemented with a B27 supplement containing insulin (ThermoFisher Scientific Inc., Waltham, MA, USA). The CMs began spontaneously contracting between dd7 and dd10. On dd10–dd12, the cells were dissociated using 0.05% trypsin and re-plated on 1:200 Geltrex-coated tissue culture plates at a density of 250,000 cells/cm^2^ to yield confluent monolayers [20]. On dd14 or dd15, the hiPSC-CMs were enriched through cultivation in glucose-free lactate supplemented DMEM medium for four days and then switched back to CMM [18]. This in vitro differentiation and selection protocol yields a preponderance of ventricular CMs [19].

The derivation and cultivation of human adipose stem cells (hASCs), differentiation of hASCs to human adipocytes (hAdips), and generation of hAdip-conditioned medium (AdCM) for co-culture or cytokine screening was performed as described [19]. Briefly, hAdips were differentiated from a single batch of isolated hASCs. The hASCs were cultured until 90% confluent. After washing the cells with DPBS containing Ca^2+^ and Mg^2+^ (ThermoFisher Scientific Inc., Waltham, MA, USA), they underwent 14 days of adipocyte differentiation to form hAdips [19]. On dd14, the hAdips were washed with DPBS and cultured in CMM, which we have previously shown could be used to culture hAdips in vitro [19]. AdCM was collected from these cells and centrifuged at 300× *g*. The supernatant was collected, filtered, and stored at −80 °C. AdCM was acquired daily for an additional 6 days.

### 2.3. Optical Mapping to Assess Membrane Voltages and Calcium Transients

From dd28 to dd30, the lactate-enriched normal- and mPKP2-derived CMs were replated onto Geltrex-coated Thermanox coverslips (ThermoFisher Scientific Inc., Waltham, MA, USA) in CMM supplemented with the ROCK inhibitor Y27632 (Tocris Bioscience, Bristol, UK). The cells were re-fed with CMM within 24 h and allowed to recover for an additional 24 h. From dd30 to dd32, the CMs were maintained in CMM (controls) or incubated with AdCM or cultured in CMM containing cytokines for 7 days with daily changes, unless otherwise indicated. The cytokines tested include monocyte chemoattractant protein-1 (100 ng/mL MCP-1, MilliporeSigma, Burlington, MA, USA), interleukins -6, -8 (100 ng/mL IL-6, 100 ng/mL IL-8; MilliporeSigma, Burlington, MA, USA), or complement factor D (100 ng/mL CFD, BioLegend, San Diego, CA, USA). All cytokines were prepared as aqueous solutions according to the manufacturer’s recommendations.

Optical mapping was performed on the hiPSC-CMs on dd34–dd38 [19]. Briefly, cell monolayers were stained with the voltage-sensitive dye di-4-ANEPPS (10 μM, ThermoFisher Scientific Inc., Waltham, MA, USA) or the calcium-sensitive dye Rhod-2AM (1 μM, ThermoFisher Scientific Inc., Waltham, MA, USA) in Tyrode’s solution (1 mM MgCl_2_, 5.4 mM KCl, 0.33 mM Na_3_PO_4_, 1.8 mM CaCl_2_, 5 mM HEPES, and 5.0 mM glucose) for 10 min at 37 °C. Prior to experimentation, the cells were washed and incubated with Tyrode’s solution lacking dye and containing blebbistatin (5 μM, Selleck Chemicals LLC, Houston, TX, USA) to eliminate motion artifacts [21]. The cells were electrically stimulated at 1 Hz with a bipolar, platinum point electrode placed near the edge of the monolayer as marked in the figures. The cell monolayers were stimulated for 1 min to reduce transient effects prior to optical recording. Imaging was performed using a CMOS camera (MiCAM Ultima, Scimedia, Costa Mesa, CA, USA) and analyzed using customized MATLAB script (Version R2021a, Mathworks, Natick, MA, USA) [21].

### 2.4. Phenotypic and Molecular Analyses

#### 2.4.1. Immunocytochemistry

Normal (JHU001) and mPKP2 (Line 398-100) CMs were plated at low confluency on 13 mm Thermanox coverslips (ThermoFisher Scientific Inc., Waltham, MA, USA) and cultured in either CMM or AdCM in 24-well plates, as described above. From dd37–dd39, the cells were fixed with 4% paraformaldehyde (ThermoFisher Scientific Inc., Waltham, MA, USA) for 10 min at room temperature (RT). Immunostaining was performed using antibodies to plakophilin-2 (MA5-18081) (ThermoFisher Scientific Inc., Waltham, MA, USA) at a dilution of 1:200, desmoplakin (25318-1-AP) (ThermoFisher Scientific Inc., Waltham, MA, USA) at 1:400, cadherin-2 (MA1-91128) (ThermoFisher Scientific Inc., Waltham, MA, USA) at 1:200, and connexin-43 (C6219) (MilliporeSigma, Burlington, MA, USA) at 1:200. Samples were fluorescently labeled with secondary antibodies (ThermoFisher Scientific Inc., Waltham, MA, USA) at a dilution of 1:200 and counterstained for nuclei using 4′,6-diamidino-2-phenylindole (DAPI, ThermoFisher Scientific Inc., Waltham, MA, USA) at 1:400 and F-actin using phalloidin (ThermoFisher Scientific Inc., Waltham, MA, USA) at a dilution of 1:400. Imaging was performed on a confocal microscope (Leica Microscystems, Wetzlar, Germany) and basic processing was conducted using open access software (ImageJ, Version 2.14.0/1.54f).

#### 2.4.2. RNA and Protein Analyses

Human iPSC-CMs were cultured in 6-well plates (ThermoFisher Scientific Inc., Waltham, MA, USA) and washed on ice in DPBS lacking Ca^2+^ and Mg^2+^ (ThermoFisher Scientific Inc., Waltham, MA, USA) prior to the isolation of RNA or protein. RNA was prepared using the RNA MiniPrep Kit (Qiagen, Hilden, Germany) according to the manufacturer’s protocol, transferred to 1.5 mL tubes, and stored at −80 °C. The concentration and purity of the isolated RNA were determined using a NanoDrop spectrophotometer (ThermoFisher Scientific Inc., Waltham, MA, USA). Reverse transcription was performed using the High-Capacity cDNA Reverse Transcription Kit (ThermoFisher Scientific Inc., Waltham, MA, USA) with 1 µg of RNA in a 20 µL reaction volume according to the manufacturer’s instructions. cDNA was quantified using the Qubit ssDNA Assay Kit (ThermoFisher Scientific Inc., Waltham, MA, USA), diluted to 2 ng/µL, and stored at −20 °C. Quantitative PCR (qPCR) was conducted using the PowerTrack SYBR Green Master Mix (ThermoFisher Scientific Inc., Waltham, MA, USA) and assayed using the ViiA 7 Real-Time PCR System (Applied Biosystems, Waltham, MA, USA). Each reaction was carried out in a 5 µL volume containing 2.5 µL of PowerTrack SYBR Green Master Mix, 0.25 µL of each forward and reverse primer, and 2 µL of cDNA template. The primers employed in this study are shown in Table 1. qPCR plates were robotically loaded on the Biomek 4000 (Beckman Coulter, Sykesville, MD, USA) and run in triplicate on all targets (see Table 1 for primers). The qPCR data were processed and analyzed in R (Version 4.3.1). Outliers were systematically determined and removed as CT values that varied from the mean of each triplicate by more than 2 units. The relative expression of the targets was determined using the 2^−ΔΔCt^ method. Two housekeeping genes, RPL32 and PPIA, were used as internal controls for normalization to ensure accurate quantification. Primer efficiencies were checked to ensure accuracy of results. Log-transformed fold change values were used for this analysis to compare the relative gene expression between conditions with an independent two-sample *t*-test for each target gene.

Protein samples were isolated on ice following the addition of radio immunoprecipitation assay (RIPA) lysis buffer (MilliporeSigma, Burlington, MA, USA) supplemented with 1X phosphatase and protease inhibitors (Cell Signaling Technology, Danvers, MA, USA). Adherent cells were dislodged using cell scrapers, and the lysate was transferred into 1.5 mL tubes. After sitting on ice for 5 min, the samples were centrifuged, and the supernatant was collected and stored at −80 °C. For Western blot analysis, equal amounts of protein determined by a BCA protein assay (ThermoFisher Scientific Inc., Waltham, MA, USA) were loaded onto SDS–PAGE gels and transferred to PVDF membranes. The membranes were blocked with 5% BSA for 1 h at RT before being incubated with primary antibodies overnight at 4 °C, including β-Actin (1:1000) (Cell Signaling Technology, Danvers, MA, USA), NF-κB p65 (1:1000) (Cell Signaling Technology, Danvers, MA, USA), phospho-NF-κB p65 (1:1000) (Cell Signaling Technology, Danvers, MA, USA), IKKβ (1:1000) (Cell Signaling Technology, Danvers, MA, USA), and GJA5 (1:1000) (Bosterbio, Pleasanton, CA, USA). After washing, the membranes were incubated with anti-mouse or anti-rabbit IgG HRP-linked antibodies (1:1000) (Cell Signaling Technology, Danvers, MA, USA) for 1 h. The membranes were then imaged using a BioRad ChemiDoc Touch Imaging System (Bio-Rad Laboratoreis, Hercules, CA, USA). The blots were quantified using ImageJ software (Version 2.14.0/1.54f) for image analysis.

### 2.5. Data and Statistical Analysis

A linear mixed model (LMM) was applied to normalize qPCR data across plates by treating each plate as a random effect in the model to account for potential plate-to-plate variability (lmer(Mean_CT~1 + (1|Plate)). Data are presented as the mean ± standard deviation (SD). Statistical tests included a one-way ANOVA with the Sidak multiple comparisons post hoc test and an independent two-sample *t*-test. Statistical differences were considered statistically significant at *p* < 0.05, and the level of significance is explicitly detailed either in the text or in the figures.

## 3. Results

### 3.1. RNA Analyses of Normal vs. PKP2 Mutant hiPSC-CMs

To evaluate paracrine-mediated effects postulated to differ between normal and mPKP2 cells, the hiPSC-CM cultures were fed daily with cardiac maintenance medium (CMM) or with human adip-conditioned medium (AdCM) for 7 days. No apparent changes in cell shape or morphology were observed during this timeframe. Quantitative differences in the RNA abundance of the normal and mPKP2 hiPSC-CMs were identified following cultivation with CMM and AdCM. Shown in Figure 1 is the relative abundance of transcripts encoding desmosome proteins, intermediate filaments, gap junctions, ATPases, and ion channels from both the normal and mPKP2 hiPSC-CMs and the responses of both cell types to AdCM relative to CMM. In experiments evaluating the normal hiPSC-CMs, we could not demonstrate any quantifiable differences in the abundance of RNAs isolated from cells cultured in CMM versus those cultured in AdCM for plakophilin-2 (PKP2), desmoplakin (DSP), connexin 43 (GJA1), the L-type calcium channel alpha-1c subunit (CACNA1C), the sarco(endo)plasmic reticulum calcium release channel (RYR2), the inward rectifying potassium channel Kir2.1 (KCNJ2), or the sodium channel protein type 5 alpha (SCN5A) (Figure 1A). However, transcripts for desmin (DES) were increased (*p* < 0.05), and transcripts encoding connexin-40 (GJA5, *p* < 0.01), the sarco(endo)plasmic reticulum calcium ATPase type 2 (SERCA2, *p* < 0.05), and the Na-K ATPase alpha-1 subunit (ATP1A1, *p* < 0.05) were significantly decreased by cultivation in AdCM compared to cultivation in CMM, suggestive of altered calcium handling (Figure 1A). In stark contrast to the normal hiPSC-CMs, the mPKP2 CMs did not show any significant changes in transcript abundance for any of the gene products analyzed, following one week of cultivation in AdCM compared to cultivation in CMM (Figure 1B). 

Since the transcript level of some genes differs between the normal and mPKP2 cells [11], we also compared the transcript levels of the normal versus mPKP2 hiPSC-CMs when both cell types were cultured for 7 days in CMM (Figure 1C) and in AdCM. No significant difference was found in the relative abundance of transcripts encoding DSP, GJA5, SERCA2, CACNA1C, RYR2, KCNJ2, or SCN5A, but transcripts for PKP2, DES, GJA1, and ATP1A1 were significantly reduced in the mPKP2 hiPSC-CMs compared to the normal hiPSC-CMs (Figure 1C). On the other hand, following 7 days of incubation in AdCM, transcripts encoding PKP2, DES, GJA1, GJA5, and KCNJ2 were reduced in the mPKP2 hiPSC-CMs relative to the normal hiPSC-CMs, while SERCA2 gene transcripts increased in abundance. No change was evident in DSP, CACNA1C, RYR2, ATP1A1, or SCN5A transcripts in the mPKP2 cells relative to the normal cells (Figure 1D). 

The data in Figure 1A–D indicate that transcript abundance differs significantly and selectively between normal and mPKP2 CMs generated from different lines of hiPSCs. The relative RNA abundance present in these cells is also modified by cultivation in CMM versus AdCM. Since the lactate-enriched CMs utilized in this study lacked the atrial-prevalent proteins encoding GJA5, our results are consistent with divergent transcriptional or post-transcriptional regulatory mechanisms innate to the ventricular CMs generated from the normal versus mPKP2 hiPSC lines. These results are also consistent with RNA levels from the PKP-mutant-derived hiPSC-CMs being less responsive to paracrine-mediated effects induced by cultivation in AdCM than the corresponding RNAs from the normal hiPSC-CMs (Figure 1A vs. Figure 1B). Since we found significant differences in RNA abundance both between the hiPSC lines differentiated into CMs and in the response of the CMs to cultivation in CMM and AdCM, all subsequent data analyses included comparisons between cell lines following incubation in either CMM or AdCM.

### 3.2. Localization of Junctional Proteins

The immunostaining of key proteins present in the ICD revealed several differences in the abundance and localization of junctional proteins in the hiPSC-CMs (Figure 1E). In the hiPSC-CM cultures with CMM, the normal cells exhibited greater staining for PKP2, cadherin-2 (CDH2), and connexin-43 (GJA1/Cx43) when compared to the mPKP2 CMs. This difference was due to the greater cytosolic staining for these proteins in the normal hiPSC-CMs. In the mPKP2 CMs, PKP2 and CDH2 proteins had a punctate, discontinuous distribution at cell junctions, while GJA1 appeared less prevalent when compared to the normal hiPSC-CMs. In contrast, DSP appeared more intensely localized to borders shared between the PKP2-mutant CMs compared to those shared between the normal cells cultured in CMM. In both lines, DSP was similarly distributed in nuclear and perinuclear spaces. The cytoskeletal actin filaments (F-actin) of the normal cells appeared to be oriented with a higher degree of parallelism than those in the mutant cells, though the thickness of these filaments appeared similar in the normal CMs compared to the PKP2-mutant CMs.

In the hiPSC-CMs maintained in AdCM, the desmosome components PKP2 and DSP showed preferential localization and finger-like projections at the cell membranes of the mPKP2 compared to the normal hiPSC-CMs. Some decrease in PKP2 and DSP cytosolic staining was also observed in the mPKP2 versus normal cells. DSP nuclear and perinuclear staining was similar in both sets of hiPSC-CMs cultured in AdCM, as well as punctate CDH2 staining at the cell membrane. GJA1 staining at the cell membrane was also punctate, but the normal hiPSC-CMs cultured in AdCM had stronger staining in the nuclear and perinuclear spaces relative to the mPKP2 cells. F-actin in the mPKP2 CMs treated with AdCM exhibited a greater degree of parallelism and appeared to align with filaments of other cells present in the syncytium, more so than the normal CMs did.

In the normal JHU001 hiPSC-CMs cultured in AdCM compared to those in CMM, PKP2 staining was more continuous along sites of cell-to-cell contact. Punctate DSP2 staining appeared more intense at the membranes of the cells cultured in AdCM relative to CMM, and under both treatments, some perinuclear staining was observed. CDH2 staining in the normal cells was continuous, with fingerlike projections in CMM, but the staining was more punctate in the normal cells cultured in AdCM. GJA1 was similarly expressed in the normal cells cultured in AdCM and in CMM, although the normal CMs in AdCM exhibited greater nuclear and perinuclear localization relative to the cells in CMM. There was no demonstratable difference in the thickness or alignment of F-actin signals in the normal hiPSC-CMs exposed to AdCM relative to CMM conditions. Likewise, in the mutant hiPSC-CMs cultured in AdCM relative to CMM, PKP2 and DSP staining were strong at the cell membranes, and the signals were distributed in a punctate manner. Nuclear and perinuclear DSP and GJA1 staining were similar between the two medium conditions. The punctate, membrane staining of CDH2 and GJA1 also were similar in the mPKP2 hiPSC-CMs cultured in both media. However, F-actin appeared more aligned with thicker filaments in the mPKP2 CMs cultured in AdCM compared to those cultivated in CMM.

### 3.3. Electrophysiology

#### 3.3.1. Voltage

We employed optical mapping to evaluate the electrophysiology of the hiPSC-CMs cultivated for 7 days in CMM or AdCM (*n* = 7), as previously described [19]. Representative traces of averaged optical transmembrane voltages and activation maps with isochrones are shown in Figure 2A and Figure 2B, respectively. We systematically evaluated the following voltage parameters: the action potential duration as the time from the maximum upstroke to 80% or 30% recovery from peak amplitude (APD_80_ or APD_30_, respectively), time to peak amplitude (T_peak_), triangulation (difference between APD_80_ and APD_30_), conduction velocity (CV), and CV heterogeneity (coefficient of variation in local CVs measured between adjacent pixels in the monolayer recording). Additional assessments of transmembrane voltages were made on the normal and mPKP2 hiPSC-CMs cultured for 24 h in CMM or AdCM.

#### 3.3.2. Voltage: Normal vs. PKP2 Mutant hiPSC-CMs Cultivated in CMM or in AdCM

In the hiPSC-CM cultures maintained in CMM for a period of 7 days, the T_peak_ and triangulation were not significantly different for the mPKP2 compared with the normal hiPSC CMs (Table 2). However, the APD_80_ (*p* < 0.05) and APD_30_ (*p* < 0.001) were significantly greater in the mPKP2 CMs, while the CV (*p* < 0.0001) and CV heterogeneity (*p* < 0.0001) were significantly less (Table 2). Because paracrine signaling can have temporal or feedback regulation [22,23], we also cultivated hiPSC-CMs in CMM and in AdCM for just 24 h. At this earlier timepoint, no significant differences were found for the APD_30_ and CV heterogeneity between the mPKP2 and normal hiPSC-CMs cultured in CMM. However, the APD_80_ (mPKP2: 321.4 ± 94.9 ms vs. normal: 224.0 ± 41.5 ms, *p* < 0.01), T_peak_ (24.0 ± 4.0 ms vs. 19.6 ± 2.5 ms, *p* < 0.05), and triangulation (136.1 ± 55.1 ms vs. 55.4 ± 8.3 ms, *p* < 0.01) were significantly elevated, while the CV was significantly reduced (6.9 ± 1.6 cm/s vs. 8.7 ± 1.4 cm/s, *p* < 0.05) in the mPKP2 cells. These results at 24 h are similar to those obtained from 7 days of culture, in which the APD_80_ was significantly elevated and the CV was significantly decreased, but differ in that the APD_30_ and CV heterogeneity did not change significantly, while the T_peak_ and triangulation were significantly higher.

In stark contrast to the changes in electrophysiology observed in CMM, at 7 days of cultivation in AdCM, we could not find a significant difference in any of the voltage parameters measured between the normal and mPKP2 hiPSC-CMs (Table 2). In the hiPSC-CMs cultured for 24 h in AdCM, the APD_30_ (mPKP2: 135.0 ± 14.0 ms vs. normal: 132.2 ± 17.6 ms) and CV (4.6 ± 0.70 cm/s vs. 5.0 ± 0.39 cm/s) again were not significantly different, but, unlike the case at 7 days of culture, the APD_80_ (393.2 ± 56.4 ms vs. 230.7 ± 55.1 ms, *p* < 0.0001), T_peak_ (28.3 ± 2.6 ms vs. 23.2 ± 1.2 ms, *p* < 0.001), triangulation (248.2 ± 24.4 ms vs. 98.6 ± 40.2 ms, *p* < 0.0001) and CV heterogeneity (0.14 ± 0.04 vs. 0.076 ± 0.012, *p* < 0.01) were significantly greater in the mPKP2 hiPSC-CMs. 

#### 3.3.3. Voltage: Normal vs. PKP2 Mutant hiPSC-CMs Cultivated in AdCM Compared to Cultivation in CMM

Next, we tested for electrophysiological differences in the hiPSC-CMs cultured in AdCM versus CMM. At 7 days of cultivation in AdCM, the normal hiPSC-CMs showed significant increases in the APD_80_ and APD_30_ relative to the CMM controls (Figure 2C), a significant decrease in the CV, but no significant change in the T_peak_, triangulation, or CV heterogeneity relative to 7 days of cultivation in CMM (Figure 2C). The mPKP2 hiPSC-CMs cultivated for the same period of time in AdCM also had a significant increase in the APD_80_ and a significant decrease in the CV relative to the CMM controls (Figure 2C), although with smaller changes in the mean values and less significant differences for the APD_80_ (mPKP2: 40.61 ms vs. normal: 60.12 ms) and CV (1.45 cm/s vs. 6.66 cm/s) (Figure 2C). Like for the normal cells, neither the triangulation, T_peak_, nor CV heterogeneity changed significantly in AdCM, although the APD_30_ was significantly greater.

In experiments with the normal hiPSC-CMs cultured in CMM for 24 h relative to AdCM for 24 h, only the CV was significantly different and reduced (Appendix A). At 7 days of culture, the CV was still significantly reduced, but the APD_80_ and APD_30_ were now significantly greater (Figure 2). In contrast, the mPKP2 hiPSC-CMs cultured for 24 h in AdCM versus CMM had significant increases in the T_peak_ and triangulation, significant decreases in the APD_30_ and CV, and no significant differences in the remaining voltage parameters (Appendix A). By 7 days of culture, the CV was still significantly decreased for cells cultured in AdCM, but so too was the APD_80_. The remaining voltage parameters were not significantly different.

#### 3.3.4. Calcium

We analyzed the calcium transients (*n* = 3) of the normal and mPKP2 hiPSC-CMs in response to changes in media. Representative traces and activation maps with isochrones are shown in Figure 3A,B. In these experiments, we systematically evaluated the following calcium parameters: the calcium transient duration as the time between calcium release and 80% or 30% recovery reuptake (CTD_80_ or CTD_30_, respectively), the decay constant that measures how quickly calcium is removed from the cytosol (Ca^2+^ decay rate), the time-to-peak Ca^2+^, the calcium conduction velocity (CV), and the CV heterogeneity (coefficient of variation in local CVs measured between adjacent pixels in the monolayer recording).

#### 3.3.5. Calcium: Normal vs. PKP2 Mutant hiPSC-CMs Cultivated in CMM or in AdCM

In experiments where the cells were cultured in CMM for 7 days, the CV, CTD_80,_ and Ca^2+^ decay rate (Table 2) did not significantly differ between the mPKP2 and normal CMs. However, the CTD_30_ (*p* < 0.001) and calcium time-to-peak (*p* < 0.0001) were significantly prolonged, while the CV heterogeneity (*p* < 0.01) was significantly reduced in the mPKP2 hiPSC-CMs (Figure 3C). In contrast, after only 24 h of cultivation in CMM, the CV (mPKP2: 9.1 ± 1.6 cm/s vs. normal: 11.5 ± 2.9 cm/s vs, *p* < 0.05) was significantly lower and the CTD_80_ (637.1 ± 63.8 ms vs. 575.3 ± 30.9 ms, *p* < 0.05) was significantly higher in the mPKP2 cells than in the normal cells.

For the hiPSC-CMs cultivated in AdCM for 7 days, no differences for the CTD_80_ could be demonstrated in the mPKP2 compared with the normal cells (Table 2). However, the time-to-peak calcium (*p* < 0.0001), CTD_30_ (*p* < 0.05), Ca^2+^ decay rate (*p* < 0.005), and CV (*p* <0.01) were significantly elevated, while the CV heterogeneity (*p* < 0.001) was significantly reduced. With just 24 h of cultivation in AdCM, the CV (mPKP2: 4.0 ± 0.5 cm/s vs. normal: 6.8 ± 1.2 cm/s, *p* < 0.0001), calcium decay rate (4.4 ± 0.2 s^−1^ vs. 4.9 ± 0.4 s^−1^, *p* < 0.001), and CTD_30_ (281.5 ± 30.6 ms vs. 336.0 ± 20.3, *p* < 0.001) decreased in the PKP2 cells compared to the normal hiPSC-CMs, unlike the results obtained after 7 days of cultivation in AdCM. It is unclear why we observed calcium transients with inconsistent patterns at 1 vs. 7 days of cultivation, but the temporal dynamics of paracrine factors may contribute to these differences.

#### 3.3.6. Calcium: Normal vs. PKP2 Mutant hiPSC-CMs Cultivated in AdCM Compared to Cultivation in CMM

We also compared the calcium transients in the normal and mPKP2 hiPSC-CMs following cultivation of each cell type in AdCM relative to cultivation in CMM (Figure 3C). Following 7 days of culture, only the CV of the normal hiPSC-CMs was significantly reduced in AdCM compared to CMM. No other measured parameter showed any significant difference between these cultivation conditions for either cell line. After just 24 h of cultivation in AdCM compared to CMM conditions, the normal hiPSC-CMs showed a significant decrease in the CV and a significant decrease in the CV heterogeneity, while the mPKP2 hiPSC-CMs showed a significant reduction in the CTD_80_, CTD_30_, T_peak_, and calcium decay rate (Appendix A).

### 3.4. Cytokines

We previously identified four immune factors (IL-6, IL-8, MCP-1, and CFD) present in human adipocyte-conditioned medium that can alter the electrophysiology of JHU001 hiPSC-CMs [19]. These cytokines were added individually in CMM to the normal and to mPKP2 hiPSC-CM monolayer cultures for 7 days with daily media changes to determine their effects on the electrophysiological properties of these cells. Activation maps with isochrones are shown in Figure 4A. In these experiments, where the normal hiPSC-CMs were cultivated only in CMM without the addition of cytokines, the APD_80_ and APD_30_ did not differ between the two sets of hiPSC-CMs; however, the CV was significantly lower (*p* < 0.001) and the triangulation was significantly higher (*p* < 0.001) in the mPKP2 compared to the normal hiPSC-CMs (Figure 4B).

Following a 7-day exposure of the normal hiPSC-CMs to each individual cytokine, the APD_80_ increased significantly in response to IL-6 but not with the other cytokines relative to the controls (CMM-only medium) (Table 3). The APD_30_ increased significantly in response to MCP-1, while the CV significantly decreased in the normal hiPSC-CMs in response to all four cytokines relative to the control conditions. These results are consistent with those observed for the normal hiPSC-CMs cultured in AdCM. On the other hand, the triangulation increased significantly in response to IL-6 and IL-8, contrary to the lack of effect of AdCM on the triangulation.

In the mPKP2 hiPSC-CMs at 7 days of incubation with individual cytokines, the APD_80_ shortened significantly in response to CFD and IL-8 but not to IL-6 or MCP-1 (Figure 4B), while the APD_30_ decreased significantly in response to (only) IL-8. These changes in the APD_30_ are contrary to the effects of AdCM on the mPKP2 cells, in which the APD_30_ increased significantly. Conversely, none of the four cytokines had a significant effect on the triangulation or CV, similar to the lack of significant effect of AdCM on these two parameters.

The directional change in the voltage parameters, excluding the CV, differed between the normal and mPKP2 hiPSC-CMs in response to the four cytokines. For example, the normal hiPSC-CMs incubated with any one of the four cytokines were generally associated with a modest prolongation in the APD_80_ relative to controls, while for the mPKP2 hiPSC-CMs, the response was always a modest shortening of the APD_80_ (Figure 4B). The APD_30_ had similar directional changes (prolongation for the normal cells and shortening for the mPKP2 cells) following the addition of any cytokines. Experimentally, we found a significant decrease in the action potential durations between the normal and mPKP2 hiPSC-CMs following treatments with CFD (APD_80_ *p* < 0.05; APD_30_ *p* < 001), IL-6 (APD_80_ *p* < 0.01; APD_30_ *p* < 0.05), IL-8 (APD_80_ *p* < 0.01; APD_30_ *p* < 0.001), and MCP-1 (APD_80_ *p* < 0.05; APD_30_ *p* < 0.01). However, no significant differences in the triangulation or in CV could be demonstrated at 7 days of incubation with the individual cytokines in the normal versus mPKP2 hiPSC-CMs.

### 3.5. NFkB

As just described, the normal versus mPKP2 hiPSC-CMs showed time-dependent differences in their responses to paracrine factors and directional responses to cytokines. Having previously observed that canonical NF-kB signaling was activated in ACM [24], we postulated that NF-kB feedback mechanisms, particularly in the mPKP2 hiPSC-CMs, might alter the effects of AdCM on these cells. To test this possibility, we evaluated total p65 (RelA), transcriptionally active phospho-p65, and total IKKβ in the cells treated for 7 days with CMM versus AdCM by Western blotting. In these experiments, we could not identify any differences in the total p65 between any of the hiPSC-CMs under either condition; however, we observed an increase in phospho-p65 and in the total IKKβ in the normal hiPSC-CMs treated with AdCM relative to CMM (Figure 5). No significant change in any of these proteins was observed in the mPKP2 hiPSC-CMs treated with AdCM.

Differential responses between the normal and mPKP2 mutant hiPSC-CMs were also observed following 7 days of treatment with IL-6, but not with any of the other cytokines tested in this study. As shown in Appendix A, a significant increase in the normalized protein levels for p65, phospho-p65, and IKKβ was observed in the normal hiPSC-CMs treated with IL-6 relative to the vehicle-treated controls. However, no significant change in any of these NF-kB associated proteins could be demonstrated in the mPKP2 hiPSC-CMs following treatment with IL-6. When we tested with MCP-1 and CFD, no significant changes in the abundance of p65, IKKβ, or phospho-p65 could be demonstrated in either the normal or mPKP2 hiPSC-CMs. These results reveal that NF-kB signaling and transcriptional responses to paracrine factors secreted by adipocytes differ between normal and mPKP2 hiPSC-CMs and may contribute to some of the molecular, cellular, and functional differences between the hiPSC-CMs examined in this study.

## 4. Discussion

In the present manuscript, we have used hiPSCs from a normal line and a mPKP2 ACM line to determine how differentiated CMs respond to paracrine factors in AdCM and to specific cytokines. We present data showing differences in the molecular (RNA), cellular (desmosome protein distribution) and functional (electrophysiology) characteristics of normal versus mPKP2 hiPSC-CMs. We identified distinct differences in the levels of selected gene transcripts and in the distribution and signal intensity of cadherin, desmosome proteins, and connexin 43 in monolayer cultures of normal versus mPKP2 hiPSC-CMs, consistent with previous reports [17]. We also showed that the phenotypic traits of normal and mPKP2 hiPSC-CM are affected both by the cultivation of CMs in AdCM compared to CMM and by the duration (1 versus 7 days) of the cultivation of these CMs in these two media. These results, together with our findings of disparate NF-kB activation, show that normal and mPKP2 hiPSC-CMs respond differentially to acute (24 hr) and chronic (7 day) activation by paracrine factors and cytokines.

The major functional differences between the hiPSC-CMs examined in this study involved APDs and the conduction velocity (CV). We determined that the baseline (control) transmembrane voltages of the hiPSC-CMs cultured in CMM differed, and that the APDs were significantly prolonged while the CV was significantly slower in the mPKP2 compared to the normal hiPSC-CMs. Following the addition of AdCM for 7 days, the increase in the APD_80_ and APD_30_, and the reduction in the CV, were greater in the normal relative to the mPKP2 hiPSC-CMs. Consequently, the CVs were not significantly different between the two groups of hiPSC-CMs cultured in AdCM. In addition to these differences between the normal and mPKP2 hiPSC-CMs, several time-dependent changes were observed. Following a 1-day incubation in AdCM relative to CMM, the APD_80_ and T_peak_ were significantly prolonged in the mPKP2 cells relative to the normal hiPSC-CMs. However, the CV was only significantly reduced in the PKP2 versus normal hiPSC-CMs cultured in CMM.

Changes to the cell electrophysiology of normal relative to mPKP2 hiPSC-CMs may affect their susceptibility to arrhythmia. At 7 days of cultivation in AdCM relative to CMM, the APD_80_ increased in both the normal and mPKP2 hiPSC-CMs, which is generally considered to be an antiarrhythmic effect [25]. The APD_30_ also significantly increased, but only in the normal hiPSC-CMs. However, after only 24 h of cultivation in AdCM compared to CMM, the APD_80_ and APD_30_ either shortened (potentially a proarrhythmic effect) or were unchanged. There was a highly significant increase in the triangulation, which also has been associated with arrhythmias [25]. The CV slowed, which can facilitate the occurrence or maintenance of reentrant arrhythmias that may contribute to the disease phenotype observed in dACM following infiltration of adipocytes [26]. At 1 and 7 days of cultivation in CMM, the CV of the mPKP2 cells was significantly lower than that of the normal hiPSC-CMs. With cultivation in AdCM for 1 and 7 days, no significant difference in the CV was found for the normal versus mPKP2 hiPSC-CMs. Hence, in mPKP2 hiPSC-CMs, the effects of 24 h, but not 7 days, of cultivation in AdCM may be proarrhythmic. Yet, given the complexity of the data across the different cell lines, additional studies will be required to identify which cytokine or combination of cytokines present in AdCM might contribute to arrhythmia in these cells.

A number of differences in calcium handling also were observed between the hiPSC-CMs cultured in CMM and in AdCM. In the mPKP2 versus normal hiPSC-CMs cultivated for 7 days in AdCM compared to CMM, very few significant changes were observed, except for a slowed CV in the normal hiPSC-CMs that was not observed in the PKP2 mutant hiPSC-CMs (Figure 3). It is unclear why we were unable to observe a similar change in the CV for calcium in the PKP2 mutant cells; however, the normal and mPKP2 hiPSC-CMs responded in a time-dependent manner to cultivation in CMM and AdCM. At 24 h in CMM, a significant prolongation in the CTD_80_ and shortening of the CV existed in the mPKP2 versus normal hiPSC-CMs, but at 7 days of cultivation in CMM, the CTD_30_, time to peak calcium, and calcium decay rates were significantly prolonged. Moreover, the CV heterogeneity was decreased. In contrast to CMM, at 24 h in AdCM, the mPKP2 mutant versus normal hiPSC-CMS had a significantly reduced CTD_30_, calcium decay rate, and CV. However, at 7 days in AdCM, significant prolongations in the CTD_30_, time to peak, CV of calcium, and calcium decay rates were observed, while the CV heterogeneity significantly decreased. The paracrine-mediated, time-dependent, abnormal calcium handling in PKP2 mutant CMs might be related to altered SERCA (Figure 1C,D) or Na/Ca exchanger (NCX1) activity, which could affect calcium transient duration times. Since dysregulated calcium handling can contribute to arrhythmias [27], these data suggest that mPKP2 hiPSC-CMs may be pro-arrhythmic and are more sensitive to acute paracrine factor activation at 24 h than normal hiPSC-CMs. However, with continued exposure for 7 days, the greater sensitivity of PKP2 mutant hiPSC-CMs to the paracrine factors is reduced.

Previously, when we evaluated the effects of the four cytokines evaluated in this study on normal hiPSC-CMs, we were able to reproduce many of the electrophysiological changes observed either after incubation with AdCM or after co-culture with hAdips [19]. Monocyte chemoattractant protein-1 (MCP-1), for example, led to a significant increase in the APD_80_ and APD_30_, and a significant decrease in the CV. Here, the responses of the normal versus mPKP2 hiPSC-CMs differed after treatments with any of the four tested cytokines: CFD, IL-6, IL-8, and MCP-1. In the normal hiPSC-CMs, these cytokines led either to prolongation or no significant difference in the APD_80_ and APD_30_, and a significantly slower CV, akin to the effects observed under AdCM exposure. Accompanying the changes in APD, the triangulation increased significantly in the normal hiPSC-CMs treated with IL-6 or IL-8, although this was not observed with AdCM. Strikingly, the functional responses of the normal and mPKP2 hiPSC-CMs after the addition of these cytokines trended in opposite directions, suggesting intrinsic differences between these two groups of CMs.

Mechanistically, the differential responses of normal versus mPKP2 hiPSC-CMs to paracrine factors and cytokines could involve (a) molecular and phenotypic differences between CMs derived from normal vs. mPKP2 hiPSC lines; (b) the altered receptor binding/sensitivity of the CMs to the paracrine factors or cytokines; or (c) the modified intracellular-signaling responses of these CMs to the activating factors. In the current study, we provide data showing that at least some of the characteristics (RNA, the spatial distribution of proteins, NF-kB signaling) of normal hiPSC-CMs differ from those of mPKP2 hiPSC-CMs. Although we did not directly measure receptor sensitivity, we did find differences in the temporal sensitivity of the cells to paracrine factors at 24 h versus 7 days, including electrophysiological differences in the hiPSC-CMs from the normal and mPKP2 cells incubated with CMM or AdCM. Chelko et al. previously reported that intracellular NF-kB signaling was activated under basal conditions in CMs derived from the same mPKP2 line used in this study [24]. Interestingly, changes in connexin 43 (GJA1, Figure 1C,D) in other systems can also influence NF-kB signaling [28]. Hence, we tested whether the cultivation of our hiPSC-CMs in AdCM leads to modified intracellular NF-kB signaling. Following 7 days of cultivation with AdCM, the normal hiPSC-CMs showed increased levels of phospho-RelA/p65 (activated NF-kB) and of IKKβ (inhibitor of NF-kB activation), but no changes in these proteins were observed in the mPKP2 cells. Similarly, IL-6 treatment of the normal but not mPKP2 hiPSC-CMs increased p65, IKKβ, and phospho-p65. Given the intrinsic NF-kB signaling present in mPKP2 hiPSC-CMs under basal conditions [24] and the paracrine mediated, time-dependent differences in function responses observed between the normal and mPKP2 hiPSC-CMs, our results suggest that the paracrine factor stimulation of mPKP2 hiPSC-CMs could lead to feedback repression of NF-kB signaling. However, we did not observe any changes in NF-kB signaling in the mPKP2 hiPSC-CMs after the addition of IL-6. Mechanistically, these data are consistent with differential NF-kB signaling between normal and mPKP2 hiPSC-CMs. Future experiments will be required to demonstrate the nuclear localization of activated NF-kB, to show direct cause–effect relationships among connexins, paracrine activation, and NF-kB signaling in these cells, and to unravel the complexity of the paracrine-mediated responses of normal versus mPKP2 hiPSC-CMs.

Several limitations in this study should be considered. First, the hiPSC-CM system depends on cells that are relatively immature [29]. Therefore, the reported results may not be fully apt to describe the behavior of more mature CMs present in adult hearts. Second, it is possible that some of the differences we observed were due to clonal line variations. Although we previously reported a good conservation of results between two normal hiPSC lines (JHU001 and WTC11) in response to AdCM [19], we have only analyzed CMs from a single clone of mPKP2 cells and compared the results with the normal JHU001 hiPSC line used in the earlier study. Ideally, either an isogenic control for the mPKP2 cell line or multiple normal and mutant PKP2 hiPSC lines would have been examined to ensure the robustness and reproducibility of the results. Third, this study focuses on the short-term (1–7 days) effects of AdCM and cytokines on the in vitro differentiation of hiPSC-CMs. In the future, it would be beneficial to investigate the long-term effects of paracrine factors on the electrophysiology of hiPSC-CMs and evaluate how the cells adapt molecularly to short- and long-term cultivation with AdCM. Fourth, we only used one batch of human hASCs to generate the hAdips. To assess batch variability, we analyzed cytokine arrays with the AdCM used in this study (unpublished data) and compared the results with our prior cytokine array data obtained from AdCM derived from different batches of hAdips [19]. In both sets of data, we found elevated levels of pro-inflammatory cytokines, including M-CSF, IL-1β, adiponectin, and MCP-1. However, newly expressed factors such as IL-11, MCP-3, and serum amyloid A (SAA) were also identified. Some of the data in the present study, regarding the normal hiPSC-CMs incubated with AdCM versus CMM, differ in significance or magnitude from those in our previous study [19]. Specifically, we observed quantitative differences in several transcripts (RYR2, CACNA1C, and KCNJ2) and differences in the magnitude and *p* values of several functional parameters (the CV heterogeneity, triangulation, CTD_80_ and CTD_30_, and calcium decay rates) [19]. Variations in cytokines present in the AdCM used in this study, thus, may contribute to the discrepancies observed with our previously published data. Some of the differences also could be attributed to the lower n value in this study for the calcium measurements and to improvements in our analysis of qPCR data that account for variability across plates run on different days. Finally, we only used one concentration of the cytokines in these studies, based on the literature of established cell lines or adult cells. It is well known that hiPSC-CMs do not always respond to drugs or small molecules in the same way that adult CMs do [30]. Despite these limitations and statistical differences, several of the main findings were reproduced between the two studies, including the significant increase in the APD_80_ and APD_30_, and significant decrease in the CV. Moreover, most of the experiments performed in the present study with the normal and mPKP2 hiPSC lines were performed in parallel instead of in series, as in our earlier study. This experimental design allows the data between these two lines to be collected at matched timepoints to minimize confounding variations in the timing of in vitro differentiation or of experimentation.

Infiltrating fat has been described as a major contributor to ventricular arrhythmias in infarcted hearts, independent of scar tissue formation [31]. In this study, we used a hiPSC-CMs model to evaluate how paracrine factors from hAdips, analogous to the infiltration of fat into the myocardium of ACM patients, may contribute to arrhythmias. Our study demonstrates that hiPSC-CMs derived from normal and mPKP2 ACM lines exhibit distinct molecular, cellular, and functional responses when exposed to paracrine factors and specific cytokines. Differences in RNA expression, desmosome protein distribution, electrophysiology, and calcium handling suggest that mPKP2 hiPSC-CMs are more prone to arrhythmic behavior, especially with acute paracrine factor exposure. Importantly, while both normal and mPKP2 hiPSC-CMs display electrophysiological alterations over time, the response to AdCM differs between the cell types. This suggests that the mechanisms underlying these variations may involve differential receptor sensitivities or altered intracellular signaling pathways such as NF-κB. These data support a role for infiltrating fat as a possible driver of arrhythmogenic propensity in ACM through the release of paracrine factors. In conclusion, our findings provide unique insights into the role of paracrine factors and cytokine-mediated signaling in arrhythmogenic cardiomyopathy, underscoring the need for further investigation to fully elucidate the drivers of these phenotypic differences.

## Figures and Tables

**Figure 1 biomedicines-12-02601-f001:**
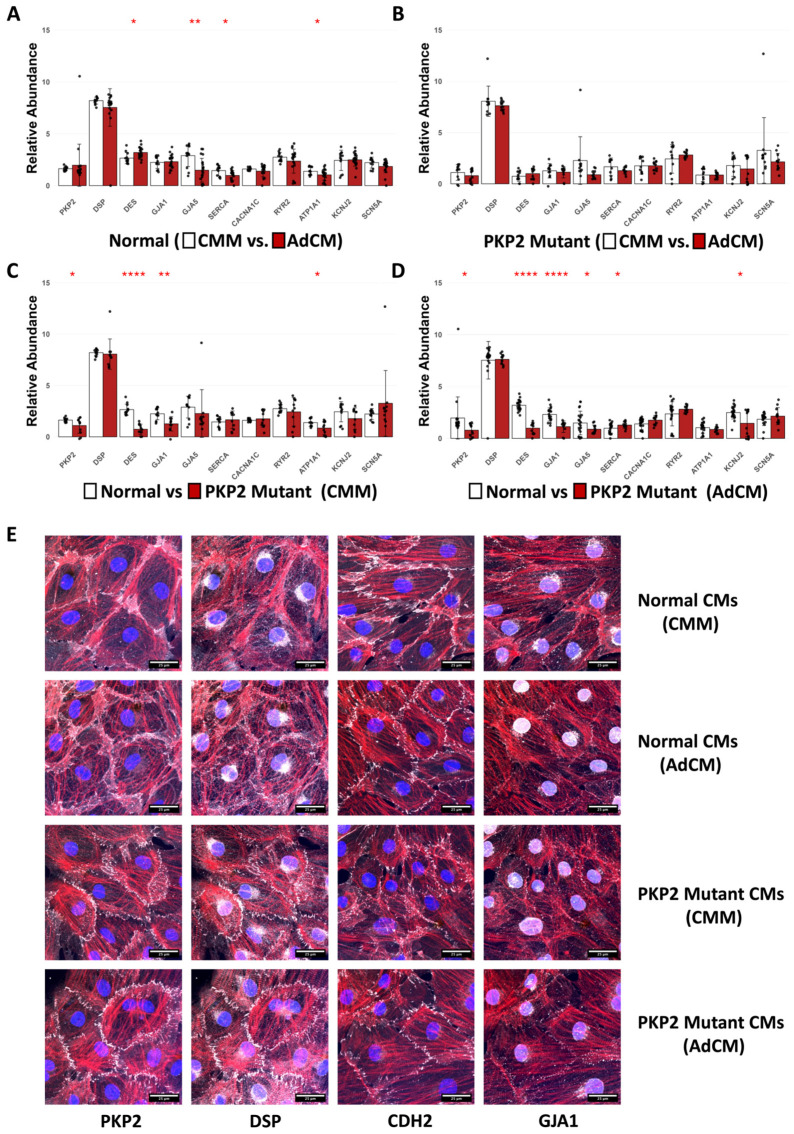
Abundance of selected gene transcripts and localization of junction proteins in normal (JHU001) and mPKP2 (398-100) hiPSC-CMs after 7 days of culture in cardiomyocyte maintenance medium (CMM) or in hAdipocyte-conditioned medium (AdCM). (**A**) Relative abundance of gene transcripts normalized to the housekeeping genes RPL32 and PPIA. Gene transcripts are expressed as a 2^−ΔΔCt^ value for normal hiPSC-CMs cultivated in CMM (left, white bars) and in cells cultured in human AdCM (right, red bars). (**B**) Normalized abundance of target genes in mPKP2 hiPSC-CMs cultivated in CMM (left, white bars) versus AdCM (right, red bars). (**C**) Normalized abundance of target genes in normal hiPSC-CMs (left, white bars) and in mPKP2 hiPSC-CMs (right, red bars) cultivated in CMM. (**D**) Normalized abundance of target genes in normal hiPSC-CMs (left, white bars) and in mPKP2 hiPSC-CMs (right, red bars), both cultivated in hAdCM. Transcripts examined include RNAs encoding cellular junctions, intermediate filaments, and ion channels. Statistical significance was assessed using independent two-sample *t*-test on four groups defined in (**A**–**D**) with significance as follows: * *p* < 0.05, ** *p* < 0.01, **** *p* < 0.0001. In group comparisons for individual gene transcripts lacking an asterisk, no significant difference (ns) could be demonstrated. (**E**) Immunostaining showing localization of plakophilin-2 (PKP2), desmoplakin (DSP), cadherin-2 (CDH2), and connexin-43 (GJA1) in monolayer cultures; signals for these proteins (white) are shown alongside F-actin (red) and nuclei (blue). Scalebar: 25 μm.

**Figure 2 biomedicines-12-02601-f002:**
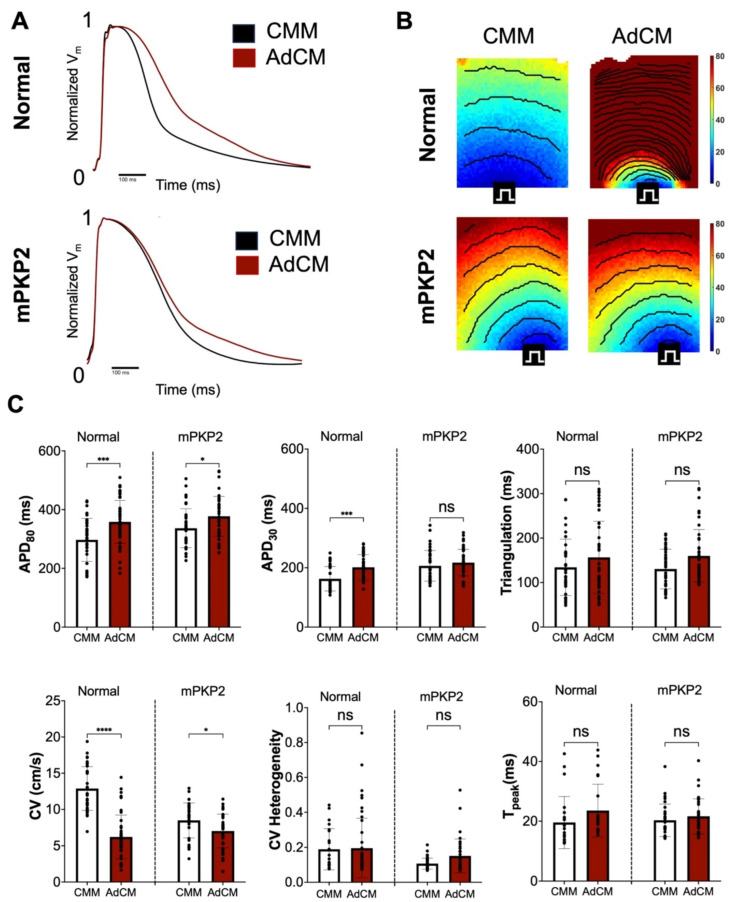
Optical voltage mapping of normal and mPKP2 hiPSC-CMs cultured in CMM or in AdCM for 7 days. (**A**) Representative traces of transmembrane voltages of hiPSC-CMs normalized from 0 to 1, at 1 Hz pacing rate. (**B**) Representative activation maps with isochrones at 10 ms intervals. (**C**) Electrophysiological characterization of the action potential durations (APD_80_ and APD_30_), triangulation, conduction velocity (CV), CV heterogeneity, and time to maximum amplitude (T_peak_) of normal and mPKP2 hiPSC-CMs incubated in CMM and AdCM as indicated in the legends. Data are expressed as mean ± SD, *n* = 7, and significance is for comparisons of cultivation in CMM vs. AdCM of normal or mPKP2 hiPSC-CMs as indicated in figure: * *p* < 0.05, *** *p* < 0.001, **** *p* < 0.0001, ns = not significant.

**Figure 3 biomedicines-12-02601-f003:**
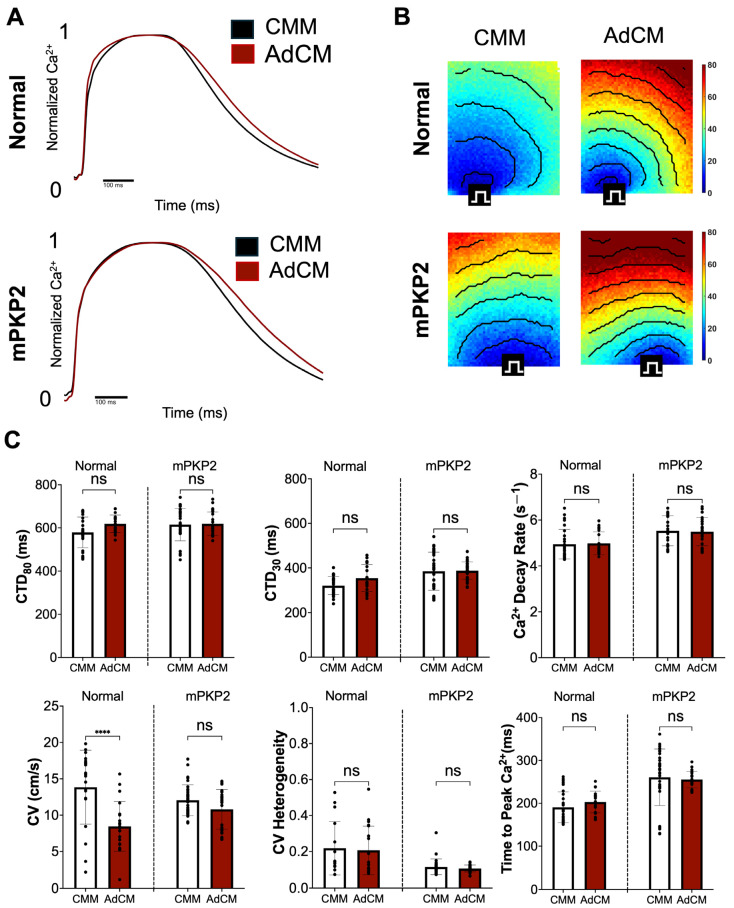
Calcium mapping of normal and mPKP2 hiPSC-CMs cultured in CMM or in AdCM for 7 days. (**A**) Representative traces of intracellular calcium transients, normalized from 0 to 1, at 1 Hz pacing rate. (**B**) Activation maps with isochrones at 10 ms intervals. (**C**) Calcium transient results, showing calcium transient duration at 80% recovery (CTD_80_), calcium transient duration at 30% recovery (CTD_30_), calcium decay rate, conduction velocity (CV), CV heterogeneity, and time-to-peak calcium. Data are expressed as mean ± SD, *n* = 4, and significance is given for comparisons of cultivation in CMM vs. AdCM of normal or mPKP2 hiPSC-CMs as indicated in figure: **** *p* < 0.0001, ns = not significant.

**Figure 4 biomedicines-12-02601-f004:**
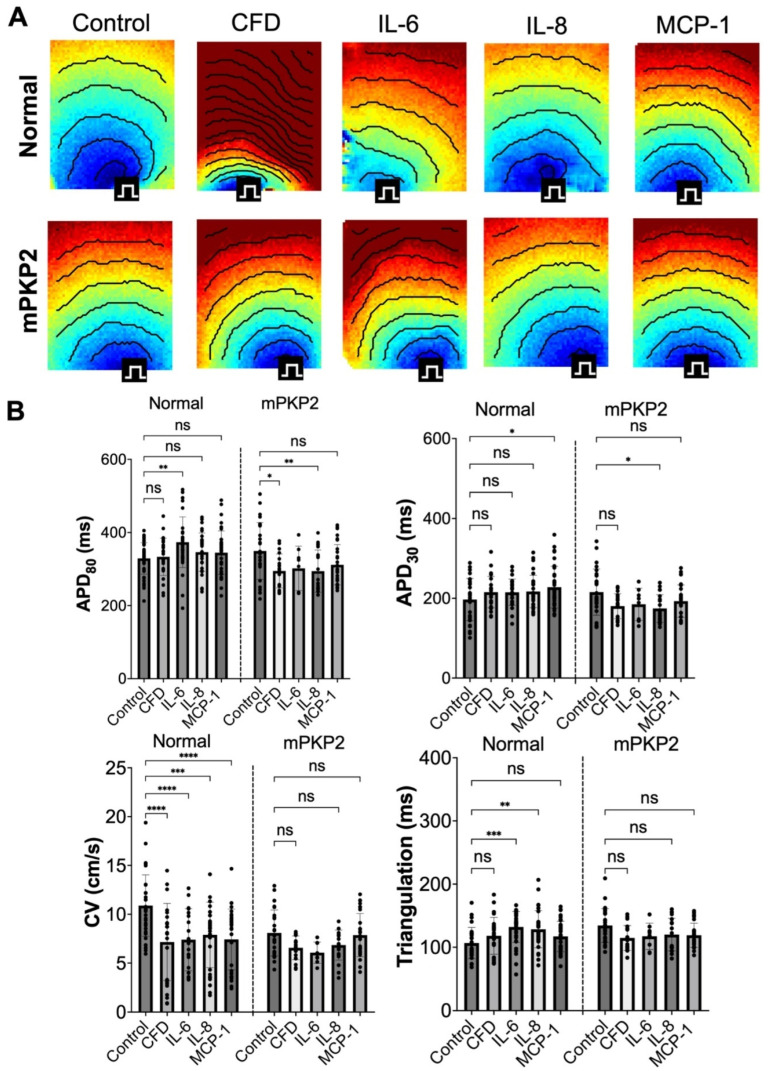
Optical mapping results of normal and mPKP2 hiPSC-CMs following addition of four cytokines to CMM for 7 days. (**A**) Activation maps with isochrones at 10 ms intervals. (**B**) Averaged datasets for APD_80_, APD_30_, CV, and triangulation for each cytokine: monocyte chemoattractant protein–1 (MCP-1), interleukin-6 (IL-6), interleukin-8 (IL-8), and complement factor D (CFD). Data are expressed as mean ± SD, with *n* = 4 and significance is for comparisons between cytokine treatments and CMM controls for normal and PKP2 mutant (mPKP2) hiPSC-CMs: * *p* < 0.05, ** *p* < 0.01, *** *p* < 0.001, **** *p* < 0.0001, ns = not significant. If line bar for significance does not appear between a group, then *n* = 2.

**Figure 5 biomedicines-12-02601-f005:**
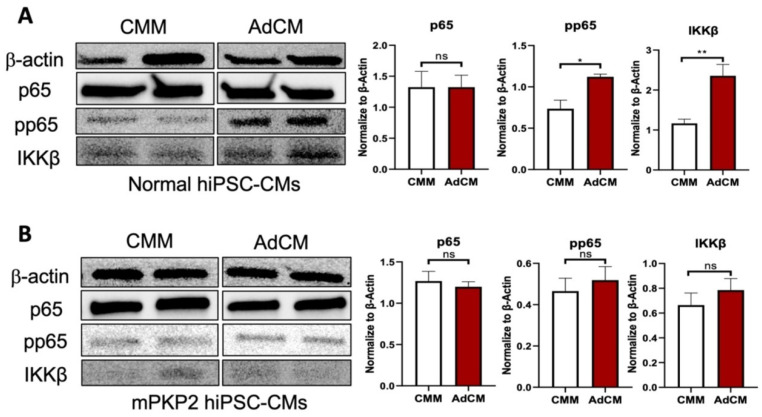
Effect of AdCM on NF-kB signaling in normal (JHU001) or mPKP2 (398-100) hiPSC-CMs. (**A**) Representative Western blots for normal hiPSC-CMs and quantified data (bar graphs) following normalization to β-actin. (**B**) Representative Western blots for mPKP2 hiPSC-CMs and quantified data following normalization to β-actin. Data are expressed as mean ± SD, *n* = 4, and *p* values as indicated: * *p* < 0.05; ** *p* < 0.01, ns = not significant. Abbreviations: RelA/p65 nuclear factor, NF-κB; pp65 (phosphorylated p65); IKKβ (IkappaB kinase); CMM (cardiomyocyte maturation medium); AdCM (adipocyte-conditioned medium).

**Table 1 biomedicines-12-02601-t001:** RNA qPCR targets and primer sequences.

Gene Transcript	Forward	Reverse
PKP2	ATGACATGCTAAAGGCTGGCA	GGGAGCTGTACTGTGCTGTTC
DSP	TCGTGCAGCCTGAATTGAAGT	CCTGGGCAAAACACTCATCC
DES	GATGAGGCAGATGCGGGAAT	CTTGAGGTGCCGGATTTCCT
GJA1	ACAGGTCTGAGTGCCTGAAC	CGAAAGGCAGACTGCTCATC
GJA5	GCAAGCACTGGGAGACGAAA	CTACCACGGTCGAGTGCTTG
SERCA	ATGGGGCTCCAACGAGTTAC	TTTCCTGCCATACACCCACAA
CACNA1C	GAAGCGGCAGCAATATGGGA	TTGGTGGCGTTGGAATCATCT
RYR2	ACAACAGAAGCTATGCTTGGC	GAGGAGTGTTCGATGACCACC
ATP1A1	ACAGACTTGAGCCGGGGATTA	TCCATTCAGGAGTAGTGGGAG
KCNJ2	GTGCGAACCAACCGCTACA	CCAGCGAATGTCCACACAC
SCN5A	GTGCCCAGAAGCAGGATGAG	GGACATACAAGGCGTTGGTG
RPL32	AGTGCCTAGTATTCTGCCAGC	AGAGTGTCTTCCAATCGCCAG
PPIA	CCCACCGTGTTCTTCGACATT	GGACCCGTATGCTTTAGGATGA

**Table 2 biomedicines-12-02601-t002:** Electrophysiological measurements of hiPSC-CMs cultured for 7 days in CMM or AdCM.

**Voltage**			
** *CMM* **			
**Parameter**	** *Normal* ** **Mean ± SD**	** *PKP2 Mutant* ** **Mean ± SD**	***p* Value**
APD_80_ (ms)	297.2 ± 73.3	336.7 ± 66.3	<0.05
APD_30_ (ms)	162.9 ± 41.2	206. 2 ± 51.8	<0.0005
Triangulation (ms)	134.3 ± 63.3	130.4 ± 44.5	Not significant (NS)
CV (cm/s)	12.9 ± 3.0	8.5 ± 2.4	<0.0001
CV Heterogeneity	0.20 ± 0.12	0.11 ± 0.03	<0.0001
T_peak_ (ms)	19.6 ± 8.7	20.3 ± 5.4	NS
** *AdCM* **			
**Parameter**	** *Normal* ** **Mean ± SD**	** *PKP2 Mutant* ** **Mean ± SD**	***p* Value**
APD_80_ (ms)	358.1 ± 73.2	377.3 ± 67.9	NS
APD_30_ (ms)	201.5 ± 42.7	217.3 ± 44.7	NS
Triangulation (ms)	156.6 ± 81.1	160.0 ± 59.2	NS
CV (cm/s)	6.2 ± 3.0	7.0 ± 2.3	NS
CV Heterogeneity	0.20 ± 0.17	0.14 ± 0.10	NS
T_peak_ (ms)	23.6 ± 8.8	21.7 ± 5.9	NS
**Calcium**			
** *CMM* **			
**Parameter**	** *Normal* ** **Mean ± SD**	** *PKP2 Mutant* ** **Mean ± SD**	***p* Value**
CTD_80_ (ms)	579.0 ± 71.1	615.4 ± 75.2	NS
CTD_30_ (ms)	320.6 ± 40.8	385.2 ± 85.4	<0.05
Ca^2+^ Decay Rate (s^−1^)	4.9 ± 0.6	5.5 ± 0.7	<0.005
CV (cm/s)	13.9 ± 5.1	12.1 ± 2.1	NS
CV Heterogeneity	0.22 ± 0.15	0.12 ± 0.04	<0.001
Time to Peak Ca^2+^ (ms)	190.7 ± 36.0	260.7 ± 66.0	<0.0001
** *AdCM* **			
**Parameter**	** *Normal* ** **Mean ± SD**	** *PKP2 Mutant* ** **Mean ± SD**	***p* Value**
CTD_80_ (ms)	618.9 ± 40.5	619.1 ± 54.4	NS
CTD_30_ (ms)	377.2 ± 60.1	388.0 ± 39.0	<0.0005
Ca^2+^ Decay Rate (s^−1^)	4.9 ± 0.6	5.5 ± 0.7	<0.005
CV (cm/s)	8.5 ± 3.4	10.8 ± 2.7	NS
CV Heterogeneity	0.21 ± 0.13	0.11 ± 0.02	<0.0005
Time to Peak Ca^2+^ (ms)	203.1 ± 25.3	255.4 ± 19.2	<0.0001

**Table 3 biomedicines-12-02601-t003:** Electrophysiological measurements of hiPSC-CMs cultured for 7 days with cytokines.

**CMM**			
**Parameter**	** *Normal* ** **Mean ± SD**	** *PKP2 Mutant* ** **Mean ± SD**	***p* Value**
APD_80_ (ms)	329.5 ± 43.2	349.6 ± 77.8	Not significant (NS)
APD_30_ (ms)	196.9 ± 53.0	215.5 ± 58.1	NS
Triangulation (ms)	106.7 ± 24.7	134.4 ± 26.8	<0.0005
CV (cm/s)	11.0 ± 3.1	8.1 ± 2.4	<0.0005
**CFD**			
**Parameter**	** *Normal* ** **Mean ± SD**	** *PKP2 Mutant* ** **Mean ± SD**	***p* Value**
APD_80_ (ms)	333.6 ± 51.1	295.0 ± 46.4	<0.05
APD_30_ (ms)	215.3 ± 39.5	180.2 ± 31.3	<0.0005
Triangulation (ms)	118.1 ± 29.1	114.8 ± 19.7	NS
CV (cm/s)	7.2 ± 4.0	6.6 ± 1.2	NS
**IL-6**			
**Parameter**	** *Normal* ** **Mean ± SD**	** *PKP2 Mutant* ** **Mean ± SD**	***p* Value**
APD_80_ (ms)	373.7 ± 69.3	302.1 ± 60.8	<0.01
APD_30_ (ms)	215.1± 33.0	184.9 ± 40.5	<0.05
Triangulation (ms)	132.1 ± 24.5	117.4 ± 20.9	NS
CV (cm/s)	7.4 ± 3.2	6.1 ± 1.1	NS
**IL-8**			
**Parameter**	** *Normal* ** **Mean ± SD**	** *PKP2 Mutant* ** **Mean ± SD**	***p* Value**
APD_80_ (ms)	346.4 ± 52.4	294.5 ± 58.0	<0.005
APD_30_ (ms)	271.1 ± 40.7	174.4 ± 34.7	<0.005
Triangulation (ms)	128.6 ± 29.3	120.1 ± 25.9	NS
CV (cm/s)	7.9 ± 3.4	6.8 ± 1.5	NS
**MCP-1**			
**Parameter**	** *Normal* ** **Mean ± SD**	** *PKP2 Mutant* ** **Mean ± SD**	***p* Value**
APD_80_ (ms)	344.9 ± 59.7	311.9 ± 55.2	<0.05
APD_30_ (ms)	227.8 ± 52.4	192.9 ± 40.7	<0.005
Triangulation (ms)	117.1 ± 24.2	119.0 ± 19.1	NS
CV (cm/s)	7.4 ± 3.1	7.9 ± 2.2	NS

## Data Availability

The data that support the findings of this study are available from the corresponding author upon reasonable request.

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
