# Peer review of "Adipocyte-Mediated Electrophysiological Remodeling of PKP-2 Mutant Human Pluripotent Stem Cell-Derived Cardiomyocytes"

_biomedicines, 2024, doi:10.3390/biomedicines12112601_

Round 1
Reviewer 1 Report
Comments and Suggestions for Authors
In the manuscript "Adipocyte-Mediated Electrophysiological Remodeling of PKP-2 mutant Human Pluripotent Stem Cell-Derived Cardiomyocytes" by Morrisse-McAlmon et al, the authors present important findings on the effects of adipocyte-derived paracrine factors on the electrophysiological properties of cardiomyocytes, with a particular focus on mPKP2 hiPSC-CMs. The study design effectively addresses the gap in understanding how adipose tissue infiltration in arrhythmogenic cardiomyopathy (ACM) affects cardiomyocyte electrophysiology, a crucial aspect of ACM pathology. However, several limitations and areas for potential enhancement can be identified.
First, as mentioned by the authors, the number of sample clones for both normal and mPKP2 hiPSC-CMs appears limited, raising concerns about the robustness and reproducibility of the results. Increasing the number of hiPSC clones, such as using mPKP2 corrected iPSCs or introducing the mPKP2 mutation into a normal iPSC clone, would strengthen the statistical power and validity of the conclusions. Additionally, while the study focuses on the short-term (1-7 days) effects of adipocyte-conditioned medium (AdCM) and cytokines, it would be beneficial to investigate the long-term effects of these factors on cardiomyocyte electrophysiology and molecular changes. Such an approach may provide a more comprehensive understanding of the progressive nature of ACM.
Furthermore, the role of NF-kB signaling in modulating the effects of adipocyte-derived factors is addressed, but the lack of detailed mechanistic investigation leaves questions unanswered. For example, the authors show data on the effects of AdCM on NF-kB signaling in normal or mPKP2 hiPSC-CMs (Figure 5), but there are no data on the effects in hiPSC-CMs treated with each cytokine. Alternatively, it might be more informative to present data on hiPSC-CMs treated with an NF-kB activator. Therefore, it remains unclear whether NF-kB activation directly mediates the electrophysiological changes or whether other pathways are involved.
In conclusion, I recommend that the authors incorporate additional data to enhance the robustness of this study and address the aforementioned limitations.
Author Response
We thank the reviewer for his/her thorough and valuable review of our submitted manuscript. In this revision, we have tried to address those comments that could be answered in a reasonable time frame. Our responses are given below.
Comment 1: First, as mentioned by the authors, the number of sample clones for both normal and mPKP2 hiPSC-CMs appears limited, raising concerns about the robustness and reproducibility of the results. Increasing the number of hiPSC clones, such as using mPKP2 corrected iPSCs or introducing the mPKP2 mutation into a normal iPSC clone, would strengthen the statistical power and validity of the conclusions.
Response #1: We agree with the reviewer's comments, but due to the amount of work necessary to generate gene edited and corrected iPSCs (which we do not have currently) and the amount of time and effort to reproduce all of the data with new lines, we believe that this is beyond the scope of the present manuscript. However have attempted to respond to your valid point. Specifically, we have added information to the Discussion to address this issue. We have edited our section on the Limitations and have written "Second, it is possible that some of the differences we observed are due to clonal line variations. Although we previously reported good conservation of results between two normal hiPSC lines (JHU001 and WTC11) in response to AdCM [19], we have only analyzed CMs from a single clone of mPKP2 cells and compared the results with the normal JHU001 hiPSC line used in the earlier study. Ideally, either an isogenic control for the mPKP2 cell line or multiple normal and mutant PKP2 hiPSC lines would have been examined to ensure the robustness and reproducibility of the results." Hopefully this addition addressing this issue will be sufficient for readers to fully understand the need for future experiments with other lines to more fully validate the results.
Comment #2: Additionally, while the study focuses on the short-term (1-7 days) effects of adipocyte-conditioned medium (AdCM) and cytokines, it would be beneficial to investigate the long-term effects of these factors on cardiomyocyte electrophysiology and molecular changes. Such an approach may provide a more comprehensive understanding of the progressive nature of ACM.
Response #2: This is an interesting point, and likely one that should be addressed. In this study our aim was to determine whether conditioned medium and associated cytokines from adipocytes might cause changes in the electrophysiology of hiPSC-CMs that might contribute to arrhythmias. We had predicted that the responses of AdCM to the PKP2 mutant cells would lead to a worsening of the electrophysiology; however, we found that the major changes at 7 days was restricted mostly to normal cells. This led to the findings that the cells have different NF-kB signaling (Western) blotting. The results are therefore quite novel. We have established a model for analyzing the effects of cytokines and AdCM on CMs from normal and ACM hiPSCs. Second, we find that the responses are unique between normal and PKP2 mutant CMs both as a function of the steady state differences between the two lines and the apparent differences in NF-kB signaling. A longer term study would be of value, but the focus then would need to be mostly on the feedback mechanisms associated with NF-kB signaling, which is outside the scope of the current manuscript. However, we have added a sentence to the discussion to address this issue. We write "...this study focuses on the short-term (1-7 days) effects of AdCM and cytokines on the in vitro differentiation hiPSC-CMs. In the future, it would be beneficial to investigate the long-term effects of paracrine factors on the electrophysiology of the hiPSC-CMs and evaluate how the cells adapt molecularly to short and long-term cultivation with AdCM."
Comment #3: Furthermore, the role of NF-kB signaling in modulating the effects of adipocyte-derived factors is addressed, but the lack of detailed mechanistic investigation leaves questions unanswered. For example, the authors show data on the effects of AdCM on NF-kB signaling in normal or mPKP2 hiPSC-CMs (Figure 5), but there are no data on the effects in hiPSC-CMs treated with each cytokine. Alternatively, it might be more informative to present data on hiPSC-CMs treated with an NF-kB activator. Therefore, it remains unclear whether NF-kB activation directly mediates the electrophysiological changes or whether other pathways are involved.
Response #3: We also strongly agree with this statement. We had in fact already begun to acquire samples for this analysis, and in response to your critique we have added new data from Western blots showing the effects of 3 cytokines on NF-kB signaling. IN these experiments, we performed the same assays that we did for the AdCM. The data are now added to the Results, a new Figure showing the results has been added to the Supplementary Data, and we have discussed our findings in the Discussion. Specifically we write:
Results: "Differential responses between normal and mPKP2 mutant hiPSC-CMs were also observed following 7 days of treatment with IL-6, but not with any of the other cytokines tested in this study. As shown in Supplemental Figure 2, a significant increase in the normalized protein levels for p65, phospho-p65, and IKKb was observed in normal hiPSC-CMs treated with IL-6 relative to vehicle treated controls. However, no significant change in any of these NF-kB associated proteins could be demonstrated in mPKP2 hiPSC-CMs following treatment with IL-6. When we tested MCP-1 and CFD, no significant changes in the abundance of p65, IKKb or phospho-p65 could be demonstrated in either normal or mPKP2 hiPSC-CMs. These results reveal that NF-kB signaling and transcriptional responses to paracrine factors secreted by adipocytes differ between normal and mPKP2 hiPSC-CMs may contribute to some of the molecular, cellular and functional differences between the hiPSC-CMs examined in this study."
Discussion: "Similarly, IL-6 treatment of normal but not mPKP2 hiPSC-CMs increased p65, IKKb and phospho-p65. Given the intrinsic NF-kB signaling present in mPKP2 hiPSC-CMs under basal conditions [24] and the paracrine mediated, time-dependent differences in function responses observed between normal and mPKP2 hiPSC-CMs, our results suggest that paracrine factor stimulation of mPKP2 hiPSC-CMs could lead to feedback repression of NF-kB signaling. However, we did not observe any changes in NF-kB signaling in mPKP2 hiPSC-CMs after the addition of IL-6. Mechanistically, these data are consistent with differential NF-kB signaling between normal and mPKP2 hiPSC-CMs. Future experiments will be required to demonstrate nuclear localization of activated NF-kB, to show a direct cause-effect relationships among connexins, paracrine activation and NF-kB signaling in these cells, and to unravel the complexity of the paracrine mediated responses of normal versus mPKP2 hiPSC-CMs."
And we refer the reviewer to the new figure including results from IL-6, which help validate our findings of altered NF-kB signaling in these cells following treatments with either AdCM or IL-6. Since the response was not seen for the other 2 cytokines, these data show that only a subset of cytokines present in adipocytes can cause the activation of NF-kB signaling - a completely new result that will likely impact our understanding of CMs and their responses to cytokines.
Comment #4: In conclusion, I recommend that the authors incorporate additional data to enhance the robustness of this study and address the aforementioned limitations.
Response #4: We thank the reviewer for his/her thoughtful/insightful comments. In the revision, we have added additional data specifically associated with NF-kB signaling in hiPSC-CMs following the additions of cytokines. These data have been added to the manuscript, and the results support our earlier findings with AdCM and strengthen the final results. While more needs to be done, particularly since some/many of the results were not as we originally predicted, we believe that the revised manuscript will be a valuable addition to the literature.
Reviewer 2 Report
Comments and Suggestions for Authors
Human iPSC technology opened a new perspective in disease model studies, mainly in genetic cardiovascular disease. It is well known that there is considerable cell line and differentiation protocol variability when electrical parameters are used to compare health and disease hiPSC-CMs. Although gene editing helped to decrease these variabilities, multi-cell line experiments are still needed to demonstrate a real physiological effect on the hiPSC-CM. Morrissette-McAlmond used a control and a PKP2-hiPSC to model arrhythmogenic cardiomyopathy with cues from adipocyte-conditioned media. The manuscript is well-written, has a good experimental design, has well-described results, and a clear discussion. Unfortunately, the results did not show as expected. The reviewer understands the challenge in creating a disease model, and probably, if the authors used a different/multiple control or mutated cells, they would have observed the expected electrophysiological results. Additionally, the cells may be at a non-responsive maturation level to the stimuli. All these comments are acknowledged in the limitation section. This study is a good starting point for future research, and I believe it should be accepted for publication after minor revisions.
- The manuscript header top is dated 2022
- Bring some of the PKP2 mutation information to the abstract section.
- Maintain consistency in describing the cells. Sometimes, there’s the cell code, and others, CTRL/PKP2.
Author Response
We would like to the thank the reviewer for their very positive critique of our submitted manuscript, and for the suggestions needed for improvement. Each comment is addressed below, and we hope with this revision that the manuscript will now be acceptable for publication.
Comment #1: The manuscript header top is dated 2022
Response #1: In the original manuscript, we did not see this date present in the header. It is likely that is occurred during the submission, as as we resubmit, we will ensure that the date is corrected.
Comment #2: Bring some of the PKP2 mutation information to the abstract section.
Response #2: Thank you for bringing this to our attention. We have now provided this information in the abstract as follows: "A normal and a PKP2 mutant (c.971_972InsT) ACM hiPSC line were cultivated and differentiated into cardiomyocytes (CMs)."
Comment #3: Maintain consistency in describing the cells. Sometimes, there’s the cell code, and others, CTRL/PKP2.
Response #3: Throughout the main text we have removed the use of the cell code (except where it was originally described), and we have replaced control with "normal" in the text where we were referring to the cells. With that said, we did leave the cell code associated with normal and mPKP2 hiPSCs to help the readers understand the origins of the cells utilized. We have done this in part in keeping with the other reviewers comments regarding the limited number of cell lines analyzed in this study.
Round 2
Reviewer 1 Report
Comments and Suggestions for Authors
The authors have tried to address the comments with additional experiments during revision, which has improved the quality and credibility of the current work.